# Duration of Steam Distillation Affects Essential Oil Fractions in Immortelle (*Helichrysum italicum*)

**Marinko Petrović** [1], **Vesna Petrović** [2], **Zdenko Mlinar** [1], **Sandra Babić** [3], **Jerko Jukić** [4], **Tatjana Prebeg** [5] **and Dario Kremer** [5,*]

1 Andrija Štampar Teaching Institute for Public Health, Department of Environmental Protection and Health Ecology, Mirogojska 16, 10000 Zagreb, Croatia; mpetrovic32@gmail.com (M.P.); zdenko.mlinar@stampar.hr (Z.M.)
2 Nicro d.o.o., Marijana Čavića 9, 10000 Zagreb, Croatia; vesna1petrovic@yahoo.com
3 Department for Analytical Chemistry, Faculty of Chemical Engineering and Technology, University of Zagreb, Trg Marka Marulića 19, 10000 Zagreb, Croatia; sbabic@fkit.hr
4 Mahovina d.o.o., Hrastje Plešivičko 12, 10450 Jastrebarsko, Croatia; jerko.jukic@gmail.com
5 Faculty of Agriculture, University of Zagreb, Svetošiminska Cesta 25, 10000 Zagreb, Croatia; tprebeg@agr.hr
\* Correspondence: dkremer@agr.hr; Tel.: +385-1-4619-422

**Abstract:** The composition of the essential oil depends on the duration and conditions under which the distillation of the plant material is carried out. In this study, one sample without fractionation and eight fractions (each after 15 min of steam distillation) of the essential oil of cultivated *H. italicum* were analysed by gas chromatography-mass spectrometry (GC-MS). The steam conditions for all samples were as follows: flow rate 800 L/h, temperature 104 °C, and pressure 0.4 bar. The test of the antimicrobial activity was performed with the modified Kirby–Bauer method (disc diffusion method) on non-selective nutrient media (blood agar) using the reference bacterial and fungal strains. A total of 75 different components were found in the essential oil samples obtained. A shorter distillation time makes the oil richer in monoterpenes and more suitable for the perfume and cosmetics industry. On the other hand, prolonged distillation leads to the essential oil being enriched with sesquiterpene oxides, which can have a negative effect on the fragrance of the essential oil. The essential oil of *H. italicum* showed antimicrobial activity only against *Staphylococcus aureus* ATCC 25923, and the best activity was shown by the sixth fraction.

**Keywords:** immortelle; essential oil; extract; fractional distillation; GC-MS; cosmetic industry

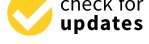



## 1. Introduction

Immortelle, or everlasting, (*Helichrysum italicum* (Roth) G. Don, family *Asteraceae*) is an evergreen, aromatic shrub that grows up to 50 cm high (Figure 1). The genus' name *Helichrysum* Mill. is derived from the Greek words *helios* (sun) and *chryos* (gold), which is due to the golden yellow colour of its flowers. After harvesting and drying, the flowers retain the same colour and shape. It is thought that this is the reason why their English names are immortelle and everlasting [1–3]. The genus *Helichrysum* comprises about 400 taxa and they are mainly distributed along the Mediterranean and grow in dry and rocky places. In southern Europe, the best-known species is *H. italicum*, which is widespread from Slovenia to Albania and Greece. In Croatia, it is widespread along the coast and on the islands. It tolerates high temperatures and drought in summer but can also survive at higher altitudes in cooler climates [4,5]. The species *H. italicum* is an erect, usually more or less tomentose, branched plant up to 50 cm tall with simple, alternate leaves. The leaves are narrowly linear, greenish, slightly tomentose to glabrescent, or sometimes white tomentose. The flowers are small and clustered in capitula aggregated in terminal corymbs up to 8 cm in diameter and surrounded by an involucre of bracts. The fruit is a cypsela (achene) [6].

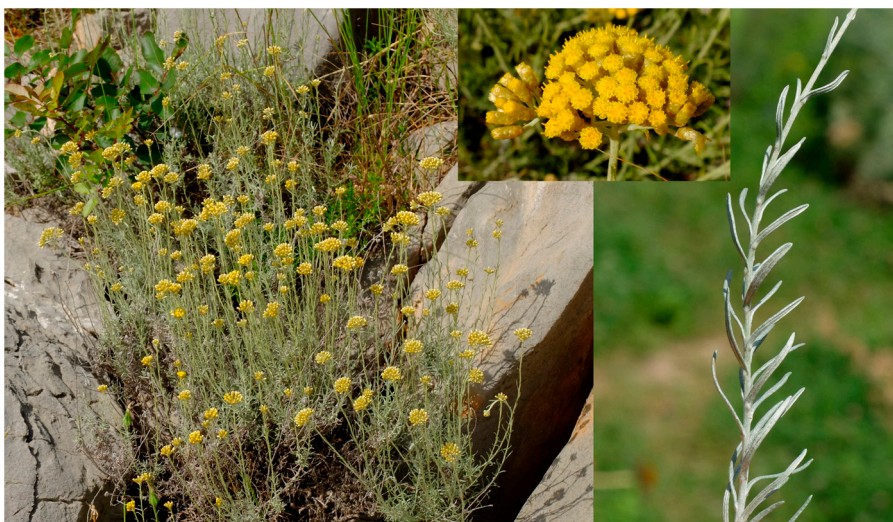

**Figure 1.** Immortelle (*Helichrysum italicum*) in natural stands with details of flowers and leaves.

The species *H. italicum* is used in traditional medicine to treat various skin infections, wounds, digestive problems, diabetes, colds, and flu. It acts as an antiseptic, diuretic, and antidepressant. Due to its regenerative effect on skin cells, it is most commonly used to treat skin diseases. It is effective against skin aging, i.e., against wrinkles and dark circles under the eyes, skin discolouration, broken capillaries, stretch marks, scars, acne, and other skin irregularities, and it soothes the skin. It is also used for dry and sensitive skin and for the treatment of psoriasis [5,7–9]. In general, the use of immortelle essential oil has increased significantly in recent years. Han et al. [10] reported that of the ten essential oils studied, only immortelle and ylang ylang (*Cananga odorata* (Lam.) Hook. f. & Thomson, family Annonaceae) inhibited the proteins associated with tissue remodelling. Several studies have shown a weaker antimicrobial activity of the essential oil of immortelle [11–15].

Due to increased application in the perfume and cosmetic industry, it has received a lot of attention, which has resulted in its excessive collection in nature. To meet the needs of the industry, *H. italicum* has been cultivated in the coastal parts of Croatia in the last few years. The wild immortelle grows in stony and dry places and commonly on poor soil [2]. In contrast, the cultivated immortelle is grown on more nutrient-rich soils and the plants are more vigorous.

The chemical composition of the essential oil of immortelle is very complex, and the characteristic odour is derived from numerous volatile compounds. Studies have shown that the essential oil of *H. italicum* contains more than 70 different components [12,16]. It can be assumed that the content of the essential oil of immortelle changes depending on the duration of distillation. Previous studies on the essential oil of *H. italicum* focused on the identification of volatile compounds in a single fraction obtained by hydrodistillation. However, there are no data on how the essential oil content of *H. italicum* changes depending on the duration of distillation. To gain insight into the relationship between the duration of steam distillation and the essential oil content, the composition of the essential oil in different fractions of the essential oil was analysed. This study was conducted in such a way that a fraction of the essential oil was extracted every 15 min during steam distillation. The eight fractions of the essential oil were obtained in this way and analysed using gas chromatography-mass spectrometry (GC-MS). In the end, the differences in oil content achieved between the individual fractions were analysed. The aim of this study is therefore to investigate the differences in the chemical composition of the essential oil fractions of *H. italicum* depending on the duration of steam distillation.

Also, only Aćimović et al. [17] have so far investigated the essential oil content of cultivated plants. Against this background, the essential oil content in cultivated plants

of *H. italicum* was analysed. Research into these two topics will expand our knowledge of immortelle as a highly sought-after plant in the cosmetics industry.

## 2. Materials and Methods

### 2.1. Plant Material

Plants of *H. italicum* were cultivated in the settlement of Miljevci in Croatia (GPS coordinates N 43°51′00″; E 16°00′46″; 220 m a.s.l.). The plantation was established by planting seedlings grown from the seeds of plants from the region of Dalmatian, Croatia. The aerial parts (stems, leaves, and flowers) of four-year-old *H. italicum* plants were harvested during the flowering period in July 2018 and identified by Dr. Dario Kremer. After harvesting, the plant parts were air-dried for 12 days protected from direct sunlight. The dried samples were stored in a dry and cool place for two weeks days before steam distillation. A sample of the plant material was deposited in the "Fran Kušan" Herbarium of the Faculty of Pharmacy and Biochemistry, University of Zagreb, Zagreb, Croatia (voucher number HFK-HR-51278).

### 2.2. Steam Distillation Conditions

The essential oil was isolated by steam distillation. The plant material of *H. italicum* (300 kg) was carefully arranged in a metal insert with a perforated bottom and placed in a stainless steel tank. The steam for the distillation was generated in a Viessmann steam generator (Viessmann, Allendorf, Germany) with a three-stage burner (Weishaupt Burner, Schwendl, Germany). The capacity of the steam generator was 600–1400 litres of steam per hour at a pressure of 0.3–0.5 bar and a temperature of 103–105 °C. After the closure, the Florentine vessel was connected to the tank. The steam was directed to the bottom of the tank at a flow rate of 800 litres per hour, a temperature of 104 °C, and a pressure of 0.4 bar. The components carried along by the steam were condensed in a water cooler and collected in a Florentine vessel. The Florentine vessel was changed every 15 min and a sample of the essential oil obtained was placed in a 10 mL glass vial. The distillation time for fractions F1, F2, F3, F4, F5, F6, F7, and F8 was 15, 30, 45, 60, 75, 90, 105, and 120 min, respectively. The distillation time for the sample without fractionation (S) was 120 min. The steam conditions for fractions F1–F8 and for the S sample were the same, i.e., flow rate 800 L/h, temperature 104 °C, and pressure 0.4 bar. Plant material from the same batch was also distilled and analysed without fractionation. The residual water in the samples was removed by adding anhydrous sodium sulphate, and the samples were stored in the refrigerator until analysis.

### 2.3. Gas Chromatography-Mass Spectrometry (GC-MS) Analyses and Compound Identification

The samples of essential oils obtained from *H. italicum* were analysed with the QP 2010 Plus GC-MS (Shimadzu, Kyoto, Japan). A capillary column ZB-5ms 60 m × 0.32 mm, with a film thickness of 0.25 μm, was used (Phenomenex, Torrance, CA, USA), and the composition of the stationary phase was 5% phenyl and 95% methyl silicone. The temperature program of the column was 60 °C (1 min), with an increase to 250 °C at a rate of 4 °C/min. The temperature of the split/split-less injector was 260 °C, and the split ratio was set to 1:10. Helium was used as the carrier gas at a flow rate of 2.2 mL/min. The temperature of the transfer line was set to 270 °C, and the detector voltage was 1,2 kV for GC-MS analyses. The samples were diluted with pentane (1:100) and injected manually (1 μL). The mass range was set to 43–350 *m/z*. Lab Solutions software version 2.72 (Shimadzu, Kyoto, Japan) was used for processing the results and quantification. The identity of the components was determined on the basis of the retention time (RT) of the components, on the basis of the GC-MS spectra and the retention indices (RI) obtained in relation to the $C_8$–$C_{20}$ n-alkanes. The AMDIS version 2.62 program was used to process the GC-MS data using the NIST library version 2.0 (both National Institute of Standards and Technology, U.S. Department of Commerce, Gaithersburg, MD, USA). The spectra and the obtained retention indices were compared with the literature [18], the in-house library, and the cited literature [19–22].

### 2.4. Antimicrobial Activity

To gain insight into the antimicrobial activity of the different fractions of the essential oil of *H. italicum*, a modified Kirby–Bauer method (disc diffusion method) was performed on non-selective culture media (blood agar). One Gram-positive strain (*Staphylococcus aureus* ATCC 25923), two Gram-negative strains (*Pseudomonas aeruginosa* ATCC 27853 and *Escherichia coli* ATCC 25922), and one strain of an opportunistic pathogenic yeast (*Candida albicans* ATCC 10231) from the stock cultures of microorganisms (Andrija Štampar Teaching Institute for Public Health, Zagreb, Croatia) were used. The blood agar culture medium was prepared by drying the medium in an incubator at 37 °C and then tempered at room temperature, after which the working suspension containing the microorganisms was inoculated onto the plate. The working suspensions were prepared separately for each strain in tubes containing 2 mL sterile saline by adjusting the suspension density to 0.5 McF (McFarland), adding Densimat (Biomerieux Biotechnology, Marcy-l'Étoile, France; control kit L1107889940). A standard of 0.5 McF corresponds to a bacterial suspension that is between $1 \times 10^8$ and $2 \times 10^8$ cfu/mL of *E. coli* [23].

After inoculation of the strains on the agar plates, sterile Whatman paper discs (Merck, Darmstadt, Germany) were soaked with 15 µL of essential oil on the surface of the inoculated agar plates. The plates were incubated under aerobic conditions at 37 °C for 24 h for bacteria or at 25 °C for 48 h for fungi. The experiment was carried out in triplicate. The diameter of the inhibition zones was measured in millimetres and expressed as mean ± standard deviation.

### 2.5. Statistical Analysis

Significant differences between the fractions for the relative percentage of compounds were determined using one-way ANOVA followed by Tukey's HSD post-hoc test ($p < 0.05$). The relationship between the individual groups of ingredients and the distillation time was evaluated using linear regression. The statistical analysis was carried out using the Statistica 7 software package (StatSoft Inc., Tulsa, OK, USA).

## 3. Results

### 3.1. GC-MS Profile

Eight fractions of *H. italicum* essential oil collected every 15 min during steam distillation were analysed by GC-MS to detect changes in essential oil composition. The chromatogram shown in Figure 2 illustrates the complexity of the composition of the essential oil of *H. italicum*. A total of 75 components were found, of which 65 were identified based on MS-spectra and RI, accounting for 81.01–92.80% of the oil. The chemical composition of the eight fractions of *H. italicum* essential oil obtained (F1–F8) and the essential oil without fractionation (S) is shown in Table 1. Unidentified components are categorised into groups based on the characteristic ions found in the spectrum.

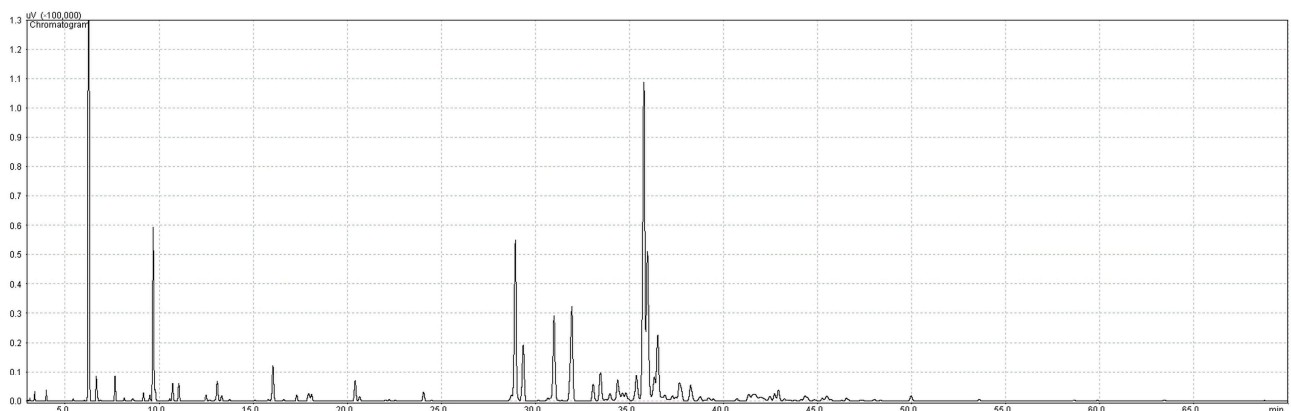

**Figure 2.** The GC-MS chromatogram of *Helichrysum italicum* essential oil without fractionation.

**Table 1.** Chemical composition of the different fractions (F1–F8) of the essential oil of *Helichrysum italicum* and the essential oil without fractionation (S), determined by GC-MS.

| Component | RT (min) | RI | Component Content (%) ± SD | | | | | | | | |
|---|---|---|---|---|---|---|---|---|---|---|---|
| | | | F1 | F2 | F3 | F4 | F5 | F6 | F7 | F8 | S |
| Methyl angelate | 3.38 | 802 | 0.14 ± 0.01 [a] | 0.12 ± 0.01 [ab] | 0.10 ± 0.01 [b] | 0.07 ± 0.01 [c] | 0.05 ± 0.01 [c] | – | 0.05 ± 0.01 [c] | – | 0.12 ± 0.01 [ab] |
| 2-Methyl-2-heptene | 3.99 | 840 | 0.16 ± 0.01 [a] | 0.11 ± 0.02 [b] | 0.08 ± 0.01 [c] | 0.07 ± 0.01 [c] | 0.06 ± 0.01 [c] | – | 0.05 ± 0.01 [c] | – | 0.13 ± 0.01 [b] |
| α-Pinene | 6.29 | 938 | 25.90 ± 0.78 [a] | 17.80 ± 0.53 [b] | 14.98 ± 0.45 [c] | 12.40 ± 0.37 [d] | 10.08 ± 0.28 [e] | 9.39 ± 0.25 [f] | 8.42 ± 0.23 [g] | 6.61 ± 0.20 [h] | 21.04 ± 0.63 [i] |
| α-Fenchene | 6.67 | 950 | 0.58 ± 0.02 [a] | 0.42 ± 0.02 [b] | 0.35 ± 0.02 [c] | 0.28 ± 0.02 [d] | 0.22 ± 0.01 [e] | 0.21 ± 0.01 [ef] | 0.19 ± 0.01 [f] | 0.15 ± 0.01 [g] | 0.46 ± 0.02 [h] |
| Camphene | 6.70 | 952 | 0.21 ± 0.01 [a] | 0.12 ± 0.01 [b] | 0.10 ± 0.01 [c] | 0.10 ± 0.01 [c] | 0.09 ± 0.01 [cd] | 0.09 ± 0.01 [cd] | 0.09 ± 0.01 [cd] | 0.07 ± 0.01 [d] | 0.18 ± 0.01 [e] |
| β-Pinene | 7.66 | 979 | 0.47 ± 0.02 [a] | 0.32 ± 0.01 [b] | 0.27 ± 0.01 [c] | 0.23 ± 0.01 [d] | 0.19 ± 0.01 [e] | 0.19 ± 0.01 [e] | 0.18 ± 0.03 [e] | 0.14 ± 0.01 [f] | 0.39 ± 0.02 [g] |
| Myrcene | 8.16 | 992 | 0.11 ± 0.01 [a] | 0.09 ± 0.01 [ab] | 0.09 ± 0.01 [abc] | 0.09 ± 0.01 [ab] | 0.08 ± 0.01 [b] | 0.10 ± 0.01 [ab] | 0.10 ± 0.01 [ab] | 0.09 ± 0.01 [ab] | 0.11 ± 0.01 [a] |
| α-Phellandrene | 8.60 | 1003 | 0.08 ± 0.01 [a] | 0.06 ± 0.01 [ab] | 0.06 ± 0.01 [ab] | 0.06 ± 0.01 [ab] | 0.05 ± 0.01 [b] | – | – | – | 0.08 ± 0.01 [a] |
| α-Terpinene | 9.18 | 1018 | 0.16 ± 0.01 [a] | 0.11 ± 0.01 [b] | 0.10 ± 0.01 [bc] | 0.09 ± 0.01 [bc] | 0.08 ± 0.02 [cd] | 0.08 ± 0.01 [cd] | 0.08 ± 0.01 [cd] | 0.06 ± 0.01 [d] | 0.13 ± 0.01 [b] |
| p-Cymene | 9.52 | 1027 | 0.16 ± 0.01 [a] | 0.12 ± 0.01 [b] | 0.11 ± 0.01 [bc] | 0.10 ± 0.01 [bcd] | 0.09 ± 0.01 [cd] | 0.09 ± 0.01 [cd] | 0.09 ± 0.01 [cd] | 0.08 ± 0.01 [d] | 0.13 ± 0.01 [b] |
| Limonene | 9.70 | 1031 | 4.55 ± 0.14 [a] | 2.99 ± 0.09 [b] | 2.58 ± 0.08 [c] | 2.37 ± 0.08 [d] | 2.05 ± 0.05 [e] | 1.91 ± 0.05 [f] | 1.86 ± 0.04 [g] | 1.52 ± 0.04 [h] | 3.56 ± 0.11 [i] |
| 1.8-Cineole | 9.78 | 1033 | 0.36 ± 0.02 [a] | 0.29 ± 0.01 [b] | 0.21 ± 0.01 [c] | 0.17 ± 0.01 [d] | 0.15 ± 0.01 [d] | 0.11 ± 0.01 [e] | 0.11 ± 0.01 [e] | 0.09 ± 0.01 [e] | 0.28 ± 0.01 [b] |
| Isobutyl angelate | 10.73 | 1054 | 0.35 ± 0.02 [a] | 0.23 ± 0.01 [b] | 0.21 ± 0.01 [bc] | 0.19 ± 0.01 [bcd] | 0.17 ± 0.01 [bcde] | 0.16 ± 0.01 [cde] | 0.16 ± 0.01 [cde] | 0.13 ± 0.01 [de] | 0.30 ± 0.02 [af] |
| γ-Terpinene | 11.06 | 1061 | 0.40 ± 0.02 [a] | 0.27 ± 0.01 [b] | 0.24 ± 0.01 [b] | 0.23 ± 0.01 [bc] | 0.19 ± 0.01 [cd] | 0.20 ± 0.01 [cd] | 0.20 ± 0.01 [cd] | 0.16 ± 0.01 [d] | 0.34 ± 0.02 [e] |
| Terpinolene | 12.51 | 1089 | 0.19 ± 0.01 [a] | 0.14 ± 0.01 [b] | 0.13 ± 0.01 [bc] | 0.12 ± 0.01 [bc] | 0.11 ± 0.01 [bc] | 0.12 ± 0.01 [bc] | 0.12 ± 0.01 [bc] | 0.10 ± 0.01 [c] | 0.17 ± 0.01 [ab] |
| Linalool | 13.09 | 1100 | 0.82 ± 0.01 [a] | 0.55 ± 0.02 [b] | 0.48 ± 0.01 [c] | 0.43 ± 0.02 [d] | 0.38 ± 0.01 [e] | 0.34 ± 0.01 [f] | 0.36 ± 0.01 [ef] | 0.29 ± 0.01 [g] | 0.65 ± 0.01 [h] |
| Isoamyl-2-Methylbutyrate | 13.33 | 1105 | 0.21 ± 0.01 [a] | 0.18 ± 0.01 [ab] | 0.17 ± 0.01 [bc] | 0.17 ± 0.01 [bc] | 0.16 ± 0.01 [bc] | 0.15 ± 0.01 [bc] | 0.16 ± 0.01 [bc] | 0.14 ± 0.01 [c] | 0.20 ± 0.01 [ab] |
| Isoamyl angelate | 16.07 | 1157 | 1.19 ± 0.03 [a] | 0.76 ± 0.02 [b] | 0.70 ± 0.02 [c] | 0.72 ± 0.03 [c] | 0.67 ± 0.02 [d] | 0.63 ± 0.02 [e] | 0.64 ± 0.02 [de] | 0.53 ± 0.02 [f] | 1.00 ± 0.03 [g] |
| Terpinene-4-ol | 17.33 | 1178 | 0.21 ± 0.01 [a] | 0.17 ± 0.01 [b] | 0.15 ± 0.01 [bc] | 0.13 ± 0.01 [cd] | 0.11 ± 0.01 [de] | 0.09 ± 0.01 [e] | 0.10 ± 0.01 [de] | 0.08 ± 0.01 [d] | 0.20 ± 0.01 [a] |
| α-Terpineol | 17.98 | 1188 | 0.48 ± 0.02 [a] | 0.44 ± 0.02 [a] | 0.45 ± 0.03 [a] | 0.49 ± 0.02 [a] | 0.51 ± 0.03 [a] | 0.58 ± 0.03 [b] | 0.62 ± 0.02 [c] | 0.63 ± 0.02 [c] | 0.48 ± 0.02 [a] |
| Myrtenol | 18.12 | 1190 | 0.26 ± 0.01 [a] | 0.27 ± 0.01 [ab] | 0.24 ± 0.01 [ac] | 0.22 ± 0.01 [b] | 0.18 ± 0.01 [c] | 0.15 ± 0.01 [d] | 0.16 ± 0.01 [d] | 0.13 ± 0.01 [e] | 0.27 ± 0.02 [ab] |
| Nerol | 20.44 | 1230 | 0.57 ± 0.02 [a] | 0.68 ± 0.03 [b] | 0.68 ± 0.03 [b] | 0.66 ± 0.02 [b] | 0.59 ± 0.02 [ad] | 0.55 ± 0.03 [ac] | 0.59 ± 0.02 [ad] | 0.51 ± 0.02 [c] | 0.63 ± 0.03 [bd] |
| MTOL | 20.68 | 1234 | – | 0.16 ± 0.01 [a] | 0.16 ± 0.01 [a] | 0.17 ± 0.01 [a] | 0.17 ± 0.01 [a] | 0.20 ± 0.01 [b] | 0.21 ± 0.01 [b] | 0.21 ± 0.01 [b] | 0.17 ± 0.01 [a] |
| Hexyl angelate | 24.07 | 1288 | 0.32 ± 0.01 [a] | 0.27 ± 0.01 [b] | 0.26 ± 0.01 [b] | 0.25 ± 0.01 [b] | 0.22 ± 0.01 [c] | 0.23 ± 0.01 [bc] | 0.23 ± 0.02 [bc] | 0.19 ± 0.01 [d] | 0.32 ± 0.01 [a] |
| Eugenol | 28.74 | 1365 | 0.37 ± 0.02 [a] | 0.32 ± 0.01 [b] | 0.23 ± 0.01 [c] | 0.26 ± 0.01 [d] | 0.28 ± 0.01 [d] | 0.21 ± 0.01 [c] | 0.23 ± 0.01 [c] | 0.20 ± 0.01 [c] | 0.27 ± 0.01 [d] |
| Neryl acetate | 29.00 | 1369 | 6.01 ± 0.18 [a] | 7.33 ± 0.23 [b] | 7.17 ± 0.21 [bc] | 7.05 ± 0.22 [c] | 6.50 ± 0.20 [d] | 6.09 ± 0.18 [a] | 6.14 ± 0.18 [a] | 5.69 ± 0.18 [e] | 6.23 ± 0.19 [a] |
| α-Copaene | 29.39 | 1375 | 3.03 ± 0.10 [a] | 2.55 ± 0.08 [b] | 2.36 ± 0.07 [c] | 2.34 ± 0.07 [c] | 2.18 ± 0.06 [d] | 2.15 ± 0.06 [d] | 2.12 ± 0.06 [d] | 1.98 ± 0.05 [e] | 2.78 ± 0.08 [f] |
| Italicene | 31.02 | 1399 [c] | 3.61 ± 0.11 [a] | 3.40 ± 0.10 [ab] | 3.26 ± 0.11 [b] | 3.18 ± 0.10 [bc] | 3.07 ± 0.10 [cd] | 3.09 ± 0.10 [cd] | 3.01 ± 0.10 [d] | 2.98 ± 0.06 [d] | 3.34 ± 0.11 [b] |
| cis-α-Bergamotene + β-Caryophyllene | 31.98 | 1416 | 5.21 ± 0.16 [a] | 5.19 ± 0.17 [a] | 5.10 ± 0.16 [ab] | 5.19 ± 0.16 [a] | 4.82 ± 0.15 [b] | 4.78 ± 0.15 [bc] | 4.71 ± 0.14 [c] | 4.57 ± 0.15 [c] | 5.24 ± 0.16 [a] |
| trans-α-Bergamotene | 33.09 | 1436 | 1.04 ± 0.03 [a] | 1.03 ± 0.03 [a] | 0.99 ± 0.02 [a] | 1.00 ± 0.03 [ab] | 0.91 ± 0.02 [b] | 0.95 ± 0.02 [ab] | 0.94 ± 0.02 [ab] | 0.91 ± 0.02 [b] | 1.05 ± 0.03 [a] |
| Italidione I | 33.51 | 1443 | 1.61 ± 0.05 [a] | 2.25 ± 0.06 [b] | 2.60 ± 0.06 [ce] | 2.84 ± 0.08 [d] | 2.86 ± 0.08 [d] | 2.69 ± 0.06 [ce] | 2.99 ± 0.08 [df] | 2.89 ± 0.08 [d] | 1.79 ± 0.08 [g] |
| α-Guaiene | 33.99 | 1451 | 0.32 ±0.01 [a] | 0.39 ±0.04 [b] | 0.39 ±0.03 [b] | 0.43 ±0.04 [bc] | 0.44 ±0.03 [c] | 0.41 ±0.04 [bc] | 0.35 ±0.03 [ab] | 0.41 ±0.03 [bc] | 0.37 ±0.03 [b] |
| α-Humulene + Neryl propionate | 34.42 | 1458 | 1.05 ± 0.02 [a] | 1.41 ± 0.03 [b] | 1.50 ± 0.05 [c] | 1.59 ± 0.04 [c] | 1.70 ± 0.06 [d] | 1.54 ± 0.04 [c] | 1.53 ± 0.04 [c] | 1.55 ± 0.04 [c] | 1.23 ± 0.04 [e] |
| α-Acoradiene | 34.65 | 1462 | 0.29 ± 0.01 [a] | 0.36 ± 0.01 [b] | 0.36 ± 0.01 [b] | 0.36 ± 0.01 [b] | 0.38 ± 0.01 [b] | 0.34 ± 0.01 [b] | 0.25 ± 0.01 [c] | 0.28 ± 0.01 [a] | 0.30 ± 0.01 [a] |

**Table 1.** *Cont.*

| Component | RT (min) | RI | Component Content (%) ± SD | | | | | | | | |
|---|---|---|---|---|---|---|---|---|---|---|---|
| | | | F1 | F2 | F3 | F4 | F5 | F6 | F7 | F8 | S |
| β-Acoradiene | 34.84 | 1465 | 0.38 ± 0.01 [a] | 0.41 ± 0.03 [a] | 0.48 ± 0.02 [b] | 0.46 ± 0.02 [b] | 0.52 ± 0.02 [b] | 0.48 ± 0.02 [b] | 0.26 ± 0.01 [c] | 0.36 ± 0.01 [a] | 0.41 ± 0.01 [a] |
| γ-Selinene | 35.41 | 1474 | 1.56 ± 0.05 [a] | 1.78 ± 0.06 [bc] | 1.87 ± 0.08 [b] | 1.70 ± 0.06 [c] | 1.98 ± 0.08 [bd] | 1.91 ± 0.07 [b] | 1.93 ± 0.07 [b] | 1.82 ± 0.06 [b] | 1.69 ± 0.06 [c] |
| γ-Curcumene | 35.88 | 1481 | 14.55 ± 0.43 [a] | 18.35 ± 0.55 [b] | 18.42 ± 0.54 [b] | 19.39 ± 0.58 [cd] | 19.88 ± 0.57 [d] | 17.58 ± 0.52 [e] | 17.24 ± 0.53 [e] | 18.81 ± 0.54 [b] | 14.89 ± 0.45 [a] |
| ar-Curcumene | 36.06 | 1484 | 8.96 ± 0.27 [a] | 10.71 ± 0.30 [b] | 11.60 ± 0.31 [cd] | 12.01 ± 0.31 [d] | 10.92 ± 0.33 [b] | 13.64 ± 0.41 [eg] | 14.16 ± 0.45 [f] | 13.33 ± 0.41 [g] | 10.91 ± 0.35 [b] |
| Italidione II | 36.42 | 1490 | 1.54 ± 0.05 [a] | 1.92 ± 0.06 [b] | 2.36 ± 0.08 [c] | 1.89 ± 0.06 [b] | 2.53 ± 0.08 [d] | 2.17 ± 0.07 [e] | 2.55 ± 0.08 [d] | 2.11 ± 0.07 [e] | 1.39 ± 0.05 [f] |
| β-Selinene | 36.60 | 1493 | 3.68 ± 0.12 [a] | 4.35 ± 0.15 [b] | 4.62 ± 0.14 [c] | 4.69 ± 0.14 [cd] | 4.95 ± 0.16 [e] | 5.06 ± 0.16 [e] | 4.84 ± 0.15 [d] | 4.71 ± 0.16 [d] | 4.02 ± 0.16 [f] |
| α-Selinene | 36.92 | 1498 | 0.35 ± 0.01 [a] | 0.44 ± 0.02 [b] | 0.56 ± 0.02 [c] | 0.43 ± 0.02 [b] | 0.68 ± 0.02 [d] | 0.61 ± 0.02 [e] | 0.51 ± 0.01 [f] | 0.42 ± 0.01 [b] | 0.31 ± 0.01 [g] |
| α-Muurolene | 37.32 | 1504 | 0.22 ± 0.01 [a] | 0.35 ± 0.01 [b] | 0.52 ± 0.02 [c] | 0.34 ± 0.01 [b] | 0.73 ± 0.02 [d] | 0.61 ± 0.03 [e] | 0.51 ± 0.02 [c] | 0.70 ± 0.02 [d] | 0.46 ± 0.02 [f] |
| β-Bisabolene | 37.49 | 1508 | – | 0.18 ± 0.01 [a] | 0.22 ± 0.01 [b] | – | 0.38 ± 0.01 [c] | 0.29 ± 0.01 [d] | 0.17 ± 0.01 [a] | 0.36 ± 0.02 [c] | 0.25 ± 0.01 [e] |
| β-Curcumene | 37.78 | 1513 | 1.20 ± 0.04 [a] | 1.84 ± 0.05 [b] | 2.14 ± 0.07 [c] | 1.89 ± 0.05 [b] | 2.26 ± 0.07 [d] | 2.21 ± 0.07 [cd] | 2.13 ± 0.06 [c] | 2.35 ± 0.07 [e] | 1.61 ± 0.06 [f] |
| γ-Cadinene | 38.30 | 1522 | 0.98 ± 0.03 [a] | 1.45 ± 0.04 [b] | 1.59 ± 0.04 [c] | 1.70 ± 0.05 [de] | 1.78 ± 0.05 [df] | 1.70 ± 0.05 [de] | 1.64 ± 0.04 [ce] | 1.84 ± 0.05 [f] | 1.28 ± 0.04 [g] |
| δ-Cadinene | 38.79 | 1531 | 0.25 ± 0.01 [a] | 0.46 ± 0.02 [b] | 0.54 ± 0.02 [c] | 0.54 ± 0.02 [c] | 0.70 ± 0.02 [d] | 0.68 ± 0.03 [d] | 0.67 ± 0.02 [d] | 0.71 ± 0.02 [d] | 0.44 ± 0.01 [b] |
| α-Cadinene | 39.23 | 1539 | 0.16 ± 0.01 [a] | 0.29 ± 0.01 [b] | 0.34 ± 0.01 [c] | 0.32 ± 0.01 [bc] | 0.50 ± 0.02 [d] | 0.51 ± 0.02 [d] | 0.44 ± 0.02 [e] | 0.48 ± 0.01 [d] | 0.29 ± 0.01 [b] |
| α-Calacorene | 39.48 | 1543 | 0.09 ± 0.01 [a] | 0.16 ± 0.01 [bc] | 0.18 ± 0.01 [c] | 0.15 ± 0.01 [b] | 0.20 ± 0.01 [c] | 0.17 ± 0.01 [bc] | 0.15 ± 0.01 [b] | 0.16 ± 0.01 [b] | 0.15 ± 0.01 [b] |
| Nerolidol | 40.74 | 1565 | 0.14 ± 0.01 [a] | 0.24 ± 0.01 [b] | 0.29 ± 0.01 [c] | 0.30 ± 0.01 [cd] | 0.32 ± 0.01 [d] | 0.50 ± 0.02 [de] | 0.47 ± 0.01 [e] | 0.47 ± 0.01 [e] | 0.25 ± 0.01 [b] |
| Neryl isovalerate | 41.38 | 1576 | 0.47 ± 0.01 [a] | 0.58 ± 0.02 [b] | 0.77 ± 0.03 [c] | 0.92 ± 0.04 [d] | 0.95 ± 0.04 [d] | 0.94 ± 0.04 [d] | 1.26 ± 0.05 [e] | 1.06 ± 0.04 [f] | 0.59 ± 0.02 [b] |
| Italidione III | 41.69 | 1581 | 1.01 ± 0.03 [a] | 1.27 ± 0.04 [b] | 1.33 ± 0.04 [b] | 1.99 ± 0.06 [c] | 2.27 ± 0.06 [d] | 2.75 ± 0.08 [e] | 2.65 ± 0.08 [e] | 3.18 ± 0.10 [f] | 1.45 ± 0.05 [g] |
| Caryophyllene oxide | 42.52 | 1595 | 0.26 ± 0.01 [ab] | 0.24 ± 0.01 [a] | 0.27 ± 0.01 [b] | 0.35 ± 0.02 [c] | 0.39 ± 0.02 [c] | 0.48 ± 0.03 [d] | 0.49 ± 0.02 [de] | 0.53 ± 0.02 [e] | 0.33 ± 0.01 [b] |
| Guaiol | 42.76 | 1599 | 0.36 ± 0.02 [a] | 0.47 ± 0.02 [b] | 0.60 ± 0.03 [c] | 0.93 ± 0.03 [d] | 1.16 ± 0.03 [e] | 1.61 ± 0.05 [f] | 1.58 ± 0.05 [f] | 2.03 ± 0.06 [g] | 0.57 ± 0.02 [c] |
| Rosifoliol | 42.98 | 1602 | 0.68 ± 0.02 [a] | 0.59 ± 0.02 [b] | 0.67 ± 0.02 [a] | 0.87 ± 0.03 [c] | 0.98 ± 0.03 [d] | 1.25 ± 0.04 [e] | 1.22 ± 0.04 [e] | 1.40 ± 0.06 [f] | 0.83 ± 0.03 [c] |
| Eudesm-5-en-11-ol | 43.28 | 1608 | 0.18 ± 0.02 [a] | 0.15 ± 0.01 [b] | 0.23 ± 0.01 [c] | 0.31 ± 0.01 [d] | 0.36 ± 0.02 [e] | 0.38 ± 0.01 [e] | 0.48 ± 0.02 [f] | 0.55 ± 0.02 [g] | 0.25 ± 0.01 [c] |
| 10-epi-γ-Eudesmol | 43.56 | 1613 | 0.09 ± 0.01 [a] | 0.05 ± 0.01 [b] | 0.11 ± 0.01 [ac] | 0.17 ± 0.01 [de] | 0.19 ± 0.01 [d] | 0.16 ± 0.01 [e] | 0.28 ± 0.01 [f] | 0.35 ± 0.01 [g] | 0.12 ± 0.01 [c] |
| SQOL1 | 44.15 | 1625 | 0.14 ± 0.01 [a] | 0.15 ± 0.01 [a] | 0.21 ± 0.01 [b] | 0.30 ± 0.01 [c] | 0.27 ± 0.01 [c] | 0.37 ± 0.02 [d] | 0.46 ± 0.02 [e] | 0.54 ± 0.01 [f] | 0.18 ± 0.01 [g] |
| SQOL2 | 44.35 | 1628 | 0.29 ± 0.01 [a] | 0.35 ± 0.02 [b] | 0.44 ± 0.01 [c] | 0.65 ± 0.02 [d] | 0.63 ± 0.02 [d] | 0.80 ± 0.03 [e] | 0.84 ± 0.03 [e] | 1.02 ± 0.04 [f] | 0.39 ± 0.02 [g] |
| γ-Eudesmol | 44.49 | 1631 | 0.25 ± 0.01 [a] | 0.19 ± 0.02 [b] | 0.22 ± 0.01 [c] | 0.25 ± 0.01 [a] | 0.34 ± 0.01 [d] | 0.45 ± 0.01 [e] | 0.49 ± 0.01 [f] | 0.52 ± 0.01 [g] | 0.32 ± 0.01 [c] |
| Epi-α-Cadinol | 44.87 | 1638 | 0.13 ± 0.01 [a] | 0.15 ± 0.01 [a] | 0.19 ± 0.01 [b] | 0.29 ± 0.02 [c] | 0.36 ± 0.01 [d] | 0.51 ± 0.02 [e] | 0.50 ± 0.02 [e] | 0.62 ± 0.02 [f] | 0.19 ± 0.01 [b] |
| β-Eudesmol | 45.29 | 1646 | 0.26 ± 0.01 [a] | 0.25 ± 0.01 [a] | 0.29 ± 0.02 [a] | 0.38 ± 0.02 [b] | 0.47 ± 0.02 [c] | 0.62 ± 0.02 [d] | 0.61 ± 0.03 [d] | 0.72 ± 0.03 [e] | 0.34 ± 0.02 [b] |
| α-Eudesmol | 45.54 | 1650 | 0.65 ± 0.03 [a] | 0.64 ± 0.03 [a] | 0.76 ± 0.03 [b] | 1.02 ± 0.05 [c] | 1.40 ± 0.07 [d] | 1.64 ± 0.06 [e] | 1.57 ± 0.05 [e] | 2.11 ± 0.07 [f] | 0.85 ± 0.03 [g] |
| SQOL3 | 45.74 | 1654 | 0.08 ± 0.01 [a] | 0.09 ± 0.01 [ab] | 0.10 ± 0.01 [bc] | 0.11 ± 0.01 [c] | – | 0.19 ± 0.01 [d] | 0.19 ± 0.01 [d] | – | 0.11 ± 0.01 [c] |
| 7-epi-α-Eudesmol | 46.54 | 1668 | 0.21 ± 0.01 [a] | 0.26 ± 0.01 [ab] | 0.33 ± 0.01 [c] | 0.48 ± 0.01 [d] | 0.61 ± 0.02 [e] | 0.31 ± 0.01 [bc] | 0.76 ± 0.02 [f] | 1.01 ± 0.03 [g] | 0.32 ± 0.01 [b] |
| β-Bisabolol | 47.30 | 1682 | – | 0.08 ± 0.01 [a] | 0.10 ± 0.01 [a] | 0.08 ± 0.01 [a] | 0.22 ± 0.01 [b] | 0.21 ± 0.01 [b] | 0.30 ± 0.01 [c] | 0.34 ± 0.01 [d] | – |
| Eudesm-7(11)-en-4-ol | 48.06 | 1695 | 0.08 ± 0.01 [ad] | 0.10 ± 0.01 [ab] | 0.12 ± 0.01 [b] | 0.17 ± 0.01 [c] | 0.19 ± 0.01 [c] | 0.06 ± 0.01 [d] | 0.24 ± 0.01 [e] | 0.27 ± 0.02 [e] | 0.12 ± 0.01 [b] |
| SQOL4 | 49.08 | 1714 | – | 0.07 ± 0.01 [a] | 0.10 ± 0.01 [b] | 0.14 ± 0.01 [c] | 0.18 ± 0.01 [d] | 0.20 ± 0.01 [de] | 0.22 ± 0.01 [e] | 0.31 ± 0.01 [f] | 0.08 ± 0.01 [a] |
| SQOL5 | 49.53 | 1723 | – | 0.07 ± 0.01 [a] | 0.08 ± 0.01 [a] | 0.12 ± 0.00 [b] | 0.13 ± 0.01 [bc] | 0.15 ± 0.01 [c] | 0.15 ± 0.01 [c] | 0.20 ± 0.01 [d] | 0.08 ± 0.01 [a] |
| SQOL6 | 49.99 | 1732 | 0.18 ± 0.02 [a] | 0.21 ± 0.01 [a] | 0.24 ± 0.01 [b] | 0.35 ± 0.01 [ch] | 0.39 ± 0.01 [d] | 0.44 ± 0.02 [e] | 0.48 ± 0.02 [f] | 0.60 ± 0.03 [g] | 0.27 ± 0.02 [h] |
| SQOL7 | 50.15 | 1735 | – | 0.06 ± 0.01 [a] | 0.07 ± 0.01 [a] | 0.10 ± 0.02 [ab] | 0.11 ± 0.01 [b] | 0.12 ± 0.01 [b] | 0.13 ± 0.02 [bc] | 0.16 ± 0.01 [c] | 0.07 ± 0.01 [a] |
| SQOL8 | 51.63 | 1763 | 0.06 ± 0.01 [a] | 0.07 ± 0.01 [a] | 0.11 ± 0.01 [b] | 0.07 ± 0.01 [a] | 0.17 ± 0.01 [c] | 0.19 ± 0.01 [c] | 0.23 ± 0.01 [d] | 0.22 ± 0.01 [d] | 0.08 ± 0.01 [a] |
| SQOL9 | 55.87 | 1846 | 0.09 ± 0.01 [ab] | 0.07 ± 0.01 [a] | 0.08 ± 0.01 [a] | 0.10 ± 0.01 [bc] | 0.13 ± 0.02 [c] | 0.17 ± 0.02 [d] | 0.16 ± 0.01 [d] | 0.21 ± 0.01 [e] | 0.12 ± 0.01 [c] |

**Table 1.** *Cont.*

| Component | RT (min) | RI | Component Content (%) ± SD | | | | | | | | |
|---|---|---|---|---|---|---|---|---|---|---|---|
| | | | F1 | F2 | F3 | F4 | F5 | F6 | F7 | F8 | S |
| Total identified | | | 99.16 | 98.70 | 98.41 | 97.89 | 97.82 | 97.17 | 96.92 | 96.52 | 98.45 |
| Monoterpene | | | 32.82 | 22.44 | 18.98 | 16.09 | 13.24 | 12.38 | 11.32 | 8.98 | 26.59 |
| Monoterpene alcohols | | | 2.35 | 2.29 | 2.16 | 2.09 | 1.93 | 1.92 | 2.04 | 1.84 | 2.40 |
| Monoterpene oxides | | | 0.36 | 0.29 | 0.21 | 0.17 | 0.15 | 0.11 | 0.11 | 0.09 | 0.28 |
| Monoterpene esters | | | 6.49 | 7.91 | 7.94 | 7.97 | 7.44 | 7.04 | 7.40 | 6.75 | 6.81 |
| Total monoterpene | | | 42.01 | 32.93 | 29.30 | 26.32 | 22.76 | 21.45 | 20.87 | 17.66 | 36.08 |
| Angelates | | | 2.01 | 1.39 | 1.28 | 1.24 | 1.12 | 1.02 | 1.08 | 0.85 | 1.72 |
| Sesquiterpene | | | 47.09 | 55.33 | 57.34 | 57.99 | 59.30 | 59.20 | 58.04 | 59.22 | 51.29 |
| Sesquiterpene alcohols | | | 3.99 | 4.30 | 5.30 | 7.23 | 8.66 | 10.34 | 11.38 | 13.74 | 5.64 |
| Sesquiterpene oxides | | | 4.15 | 5.45 | 6.30 | 6.73 | 7.66 | 7.65 | 8.19 | 8.19 | 4.64 |
| Total sesquiterpene | | | 55.23 | 65.09 | 68.94 | 71.96 | 75.63 | 77.17 | 77.61 | 81.15 | 61.57 |
| Other compounds | | | 0.75 | 0.60 | 0.48 | 0.49 | 0.49 | 0.37 | 0.44 | 0.34 | 0.61 |

RT, retention time; RI, retention indices. F1–F8, fractions obtained during steam distillation by separation every 15 min; S, sample obtained without fractionation; MTO, unidentified monoterpene alcohol; SQOL1–SQOL9, unidentified sesquiterpene alcohols; C, content; SD, standard deviation; superscript letters indicate a statistically significant difference between fraction at $p < 0.05$.

### 3.2. The Association between Distillation Duration and Chemical Degradation

The results obtained indicate significant changes in the composition of the essential oil during steam distillation. The greatest difference between the individual components was found for α-pinene, the proportion of which fell from 25.90% (fraction F1) to 6.61% (fraction F8). It is followed by limonene with a decrease from 4.55% to 1.52% (Table 1). Both compounds are monoterpenes, and their content decreases the most during distillation. The content of angelate also decreased during distillation. These two groups of components are more volatile and it is understandable that the first fractions contain more of them.

Sesquiterpenes and sesquiterpene oxides are less volatile, so their content increases towards the end of distillation. The most common sesquiterpenes were γ-curcumene, ar-curcumene, and β-selinene. The content of the predominant sesquiterpene γ-curcumene increased from 14.55% (fraction F1) to 18.81% (fraction F8) with the highest content (19.88%) in fraction F5 (Table 1). The content of the more volatile sesquiterpenes changed less in the different fractions. The content of italicene decreased slightly from 3.61% to 2.98%. It is also obvious that the amount of neryl acetate varied during distillation. The highest content of neryl acetate (7.33%) was found in fraction F2, and the lowest (5.69%) in fraction F8. This is consistent with the volatility of neryl acetate, which lies between monoterpenes and sesquiterpenes.

The relationship between the component groups and the distillation time is shown in Figure 3. The coefficients of determination show that the changes in the proportion of individual components and groups of components during distillation are not linear. The coefficient of determination ($R^2$) for α-pinene was 0.9544, for italicenes 0.9239, and for total italidones 0.9220. The coefficient of determination for monoterpenes, monoterpene esters, and sesquiterpenes was 0.9043, 0.9461, and 0.9220, respectively. The coefficient of determination without the first fraction for the total sesquiterpenes was 0.9006.

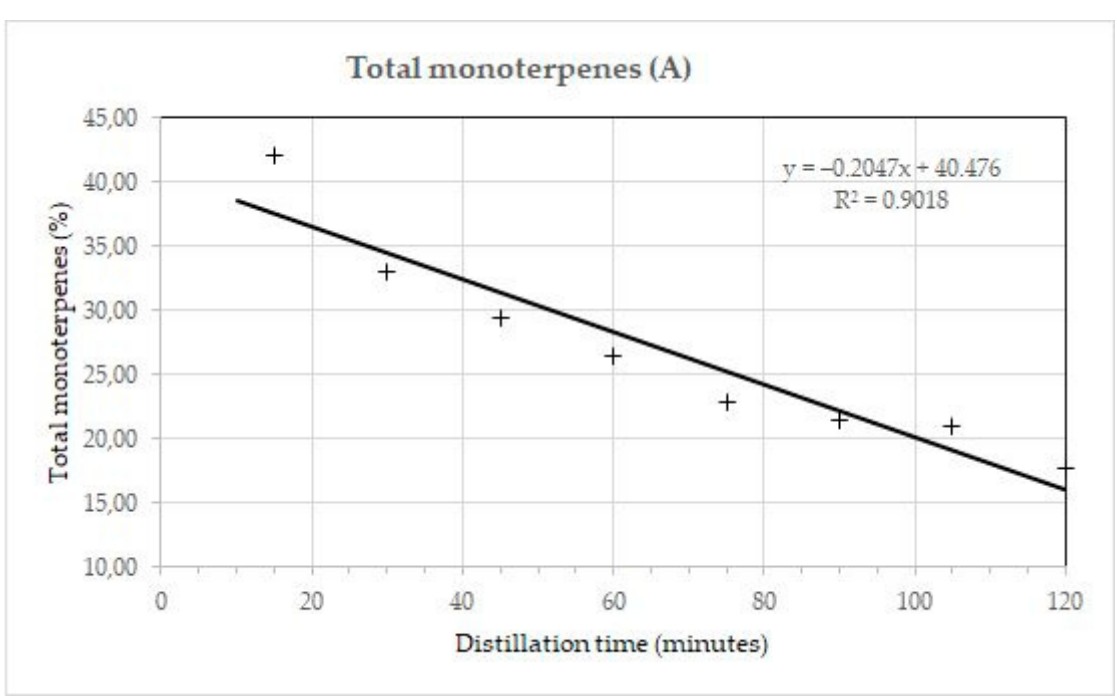

**Figure 3.** *Cont.*

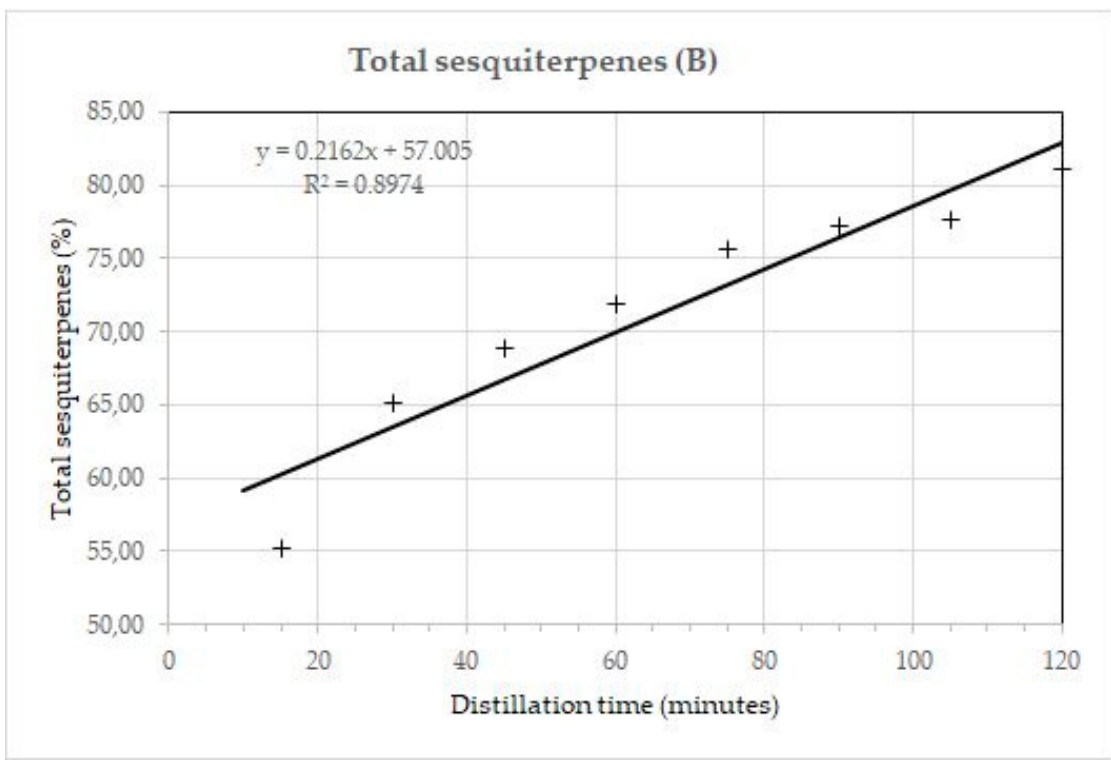

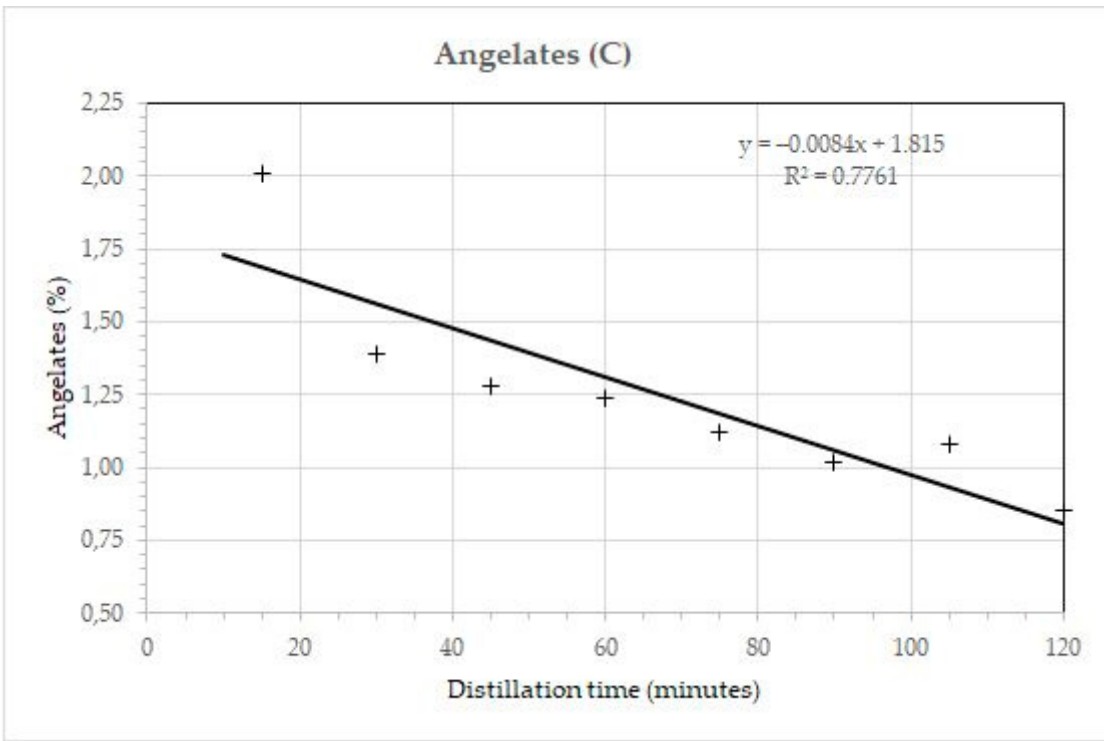

**Figure 3.** *Cont*.

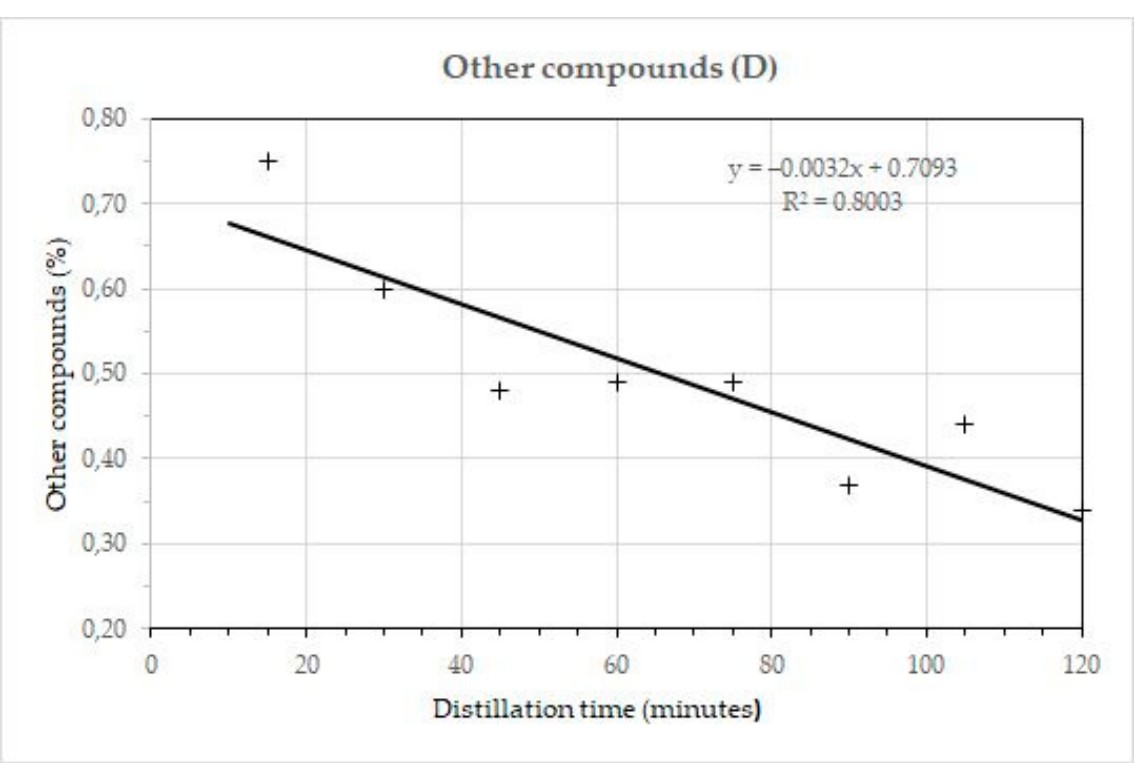

**Figure 3.** Dependence of total monoterpenes (**A**), total sesquiterpenes (**B**), angelates (**C**), and other compounds (**D**) in essential oil *Helichrysum italicum* on the duration of steam distillation. +, percentage of components in a particular fraction.

### 3.3. Antimicrobial Activity

In our study, we also tested the antimicrobial activity of the essential oil of *H. italicum* to gain a first insight into the possible differences between the fractions (Figure 4). The results showed antimicrobial activity of the essential oil only against *Staphylococcus aureus* ATCC 25923 without significant differences between the fractions (Table 2). There is no activity against *Pseudomonas aeruginosa* ATCC 27853, *Escherichia coli* ATCC 25922, and *Candida albicans* ATCC 10231.

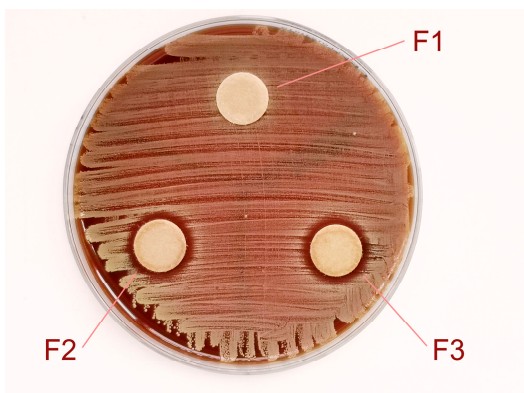

**Figure 4.** Antimicrobial activity of *H. italicum* essential oil on *Staphylococcus aureus* ATCC 25923. Fraction F1 without wide inhibition zone and fractions F2 and F3 with clear inhibition zones.

**Table 2.** Antimicrobial activity of the essential oil of *Helichrysum italicum*. Inhibition zone (mm) of the essential oil of *Helichrysum italicum*, expressed as the mean of three independent measurements.

| ATCC Strain | F1 IZ ± SD | F2 IZ ± SD | F3 IZ ± SD | F4 IZ ± SD | F5 IZ ± SD | F6 IZ ± SD | F7 IZ ± SD | F8 IZ ± SD | S IZ ± SD |
|---|---|---|---|---|---|---|---|---|---|
| *Staphylococcus aureus* ATCC 25923 | – | 11 ± 2.0 | 12 ± 1.0 | 12 ± 1.5 | 11 ± 1.0 | 13 ± 1.0 | 10 ± 1.0 | 11 ± 2.0 | 9 ± 0.5 |
| *Pseudomonas aeruginosa* ATCC 27853 | – | – | – | – | – | – | – | – | – |
| *Escherichia coli* ATCC 25922 | – | – | – | – | – | – | – | – | – |
| *Candida albicans* ATCC 10231 | – | – | – | – | – | – | – | – | – |

F1–F8, fractions obtained during steam distillation by separation every 15 min; S, sample obtained without fractionation; IZ, inhibition zone; SD, standard deviation.

## 4. Discussion

The chemical composition of eight fractions of the essential oil *H. italicum* and the essential oil without fractionation was analysed. A total of 65 components were identified using MS-spectra and RI. Several authors have studied the composition of the essential oils of *H. italicum,* and most have identified over 40 components [1,2,19–22,24]. In several papers, α-pinene was also the most abundant monoterpene, while neryl acetate was the most abundant monoterpene ester. Greater differences in the composition of the sesquiterpene fraction of the essential oil of *H. italicum* were found by Kladar et al. [1], Morone-Fortunato et al. [19], and Melito et al. [20]. Kladar et al. [1] investigated the essential oil of *H. italicum* from a wild population from Montenegro and found that the content of neryl acetate, γ-curcumene, and *ar*-curcumene was 28.2%, 7.8%, and 8.3%, respectively. On the other hand, the proportion of the monoterpene α-pinene was only 0.4%. Morone-Fortunato et al. [19] analysed the influence of the genotypes of *H. italicum* on the essential oil composition and found that the neryl acetate content varied between 0.7% and 32.0%, γ-curcumene between 0.3% and 41.0%, and *ar*-curcumene between 1.1% and 8.3%, depending on the genotype. In the same study, the α-pinene content was low and even below the quantification limit in some genotypes [19]. According to Mastelić et al. [24], α-pinene was the most abundant compound in the essential oil of the Croatian population of *H. italicum*, accounting for 12.8%. Blažević et al. [25] also studied wild populations of *H. italicum* from Croatia and showed that the main constituent of the essential oil before flowering was α-pinene (28.86%). The same authors found that the amount of α-pinene decreases with increasing flowering, reaching only 4.15% during the flowering period. Before flowering, the γ-curcumene content was 16.65%, and during flowering it dropped to 12%. In contrast, the α-curcumene content was lower before flowering (0.44%), while it amounted to 28.06% during flowering [25]. The samples of two Croatian populations examined by Malenica Staver et al. [14] showed a slightly lower content of α-pinene (21.6%). Čavar Zeljković et al. [16] found that the major sesquiterpene components in the essential oil of *H. italicum* from wild populations in Croatia are α-*trans*-bergamotene (10.2%) and β-acoradiene (10.1%). These components were present in our essential oil in amounts of about 1% and 0.5%, respectively. According to Odak et al. [26], the essential oil of *H. italicum* from Bosnia and Herzegovina also contains a large amount of α-pinene (29.4%) and limonene (5.2%). All the above-mentioned articles deal with wild populations of *H. italicum*. On the other hand, Aćimović et al. [17] analysed the essential oil of cultivated plants. They found that the main constituents of the essential oil obtained from plants cultivated in Serbia were γ-curcumene (13.6%), β-selinene (12.2%), *a*-pinene (11.8%), β–caryophyllene (6.7%), and neryl acetate (5.5%). The observed differences can be attributed to different genotypes as well as different climatic and soil conditions between the populations studied. Similar variations are known for some other plants. Turner et al. [27], for example, found that *Apium graveolens* (Apiaceae) from 2018 contained a higher level of sesquiterpenes due to the higher temperatures in that year.

Our study has shown that the duration of distillation had a significant influence on the composition of the essential oil of *H. italicum*. The use of milder distillation conditions or a

shorter distillation time makes the oil richer in monoterpenes. If the distillation period is too long, the oil will become enriched with sesquiterpene oxides, which can have a negative effect on the fragrance of the essential oil. Andreani et al. attempted to determine the compounds responsible for the fragrance of the essential oil of *H. italicum* [22]. They found that the most odor-active constituents in the *H. italicum* essential oil are 4,6-dimethyloctane-3,5-dione, 1,8-cineole, and nerol. The 4,6-dimethyloctane-3,5-dione was not determined in our study. The content of the monoterpene 1,8-cineole, also known as eucalyptol, ranged from 0.36% in the F1 fraction to 0.09% in the F8 fraction (Table 1). In contrast, the content of the monoterpenoid alcohol nerol varied slightly between the fractions, decreasing from 0.68 in fractions F2 and F3 to 0.51 in fraction F8. Given that the largest amount of cineol and nerol is found in the first fractions, it can be concluded that the first fractions are the most suitable for application in the perfume industry. With this knowledge, it may be possible for some groups of cosmetic products to shorten the distillation time and thus achieve certain energy savings.

The essential oils of many aromatic plants have biological activity, including antimicrobial activity. In our study, the essential oil *H. italicum* showed antimicrobial activity only against *Staphylococcus aureus* ATCC 25923, with no significant differences between the fractions (Table 2). The antimicrobial reaction to the F1 sample showed no visible zone, although no growth of microorganisms was detected under the disk. Why no additional inhibition occurred in this sample remains a question that needs to be answered. The fact that the growth inhibition is pronounced in the Gram-positive bacterial strain is probably related to the somewhat simpler structure of the bacterial cell. The fact is that Gram-positive bacterial cells have no outer membrane and, relatively speaking, are surrounded by a simpler peptidoglycan layer. This peptidoglycan layer is several times thicker than the multiple peptidoglycan layers of Gram-negative bacteria, which of course also have an additional outer membrane [28]. The outer membrane is composed of glycolipids (mainly lipopolysaccharide), i.e., a complex compound of lipids, sugars, and proteins, which makes it somewhat more resistant to the influence of the essential oil of *H. italicum*, especially compared to Gram-positive microorganisms [29,30]. The function of lipopolysaccharide is that it serves as a barrier to various external chemicals and enzymes that could damage the cell. For this reason, Gram-negative bacteria are better able to protect themselves from damage caused by essential oils, but this is also due to a number of additional mechanisms that help them deal with DNA, protein, and membrane damage.

Bouzid et al. [13] found that the essential oil of *H. italicum* had the strongest inhibitory effect on *Staphylococcus aureus* and *Micrococcus luteus*. Haiying et al. [31] tested the antibacterial activity of the essential *H. italicum* oil on seven different microorganisms and found that the strongest inhibitory effect was against *Staphylococcus aureus* and *Escherichia coli*. According to Malenica Staver et al. [14], the essential oil of H. italicum showed a weak inhibitory effect on Gram (+) bacteria (*Staphylococcus aureus* and *S. epidermidis*) and *Candida albicans* with an inhibition zone of 7.5 to 8.5 mm, but there was no inhibitory activity against Gram (−) bacteria (*Pseudomonas aeruginosa* and *Escherichia coli*). Furthermore, the results of the microdilution assay showed that the essential oil of *H. italicum* has a bacteriostatic effect on *Staphylococcus aureus* and Acinetobacter *baumannii* ATCC BAA-1605 [14]. Kutluk et al. [14] found that H. italicum essential also has antiviral activity against DNA and RNA viruses. Extensive studies by Aćimović et al. [17] on seven Gram (−) bacteria and nine Gram (+) bacteria showed that the essential oil of *H. italicum* showed little or no activity against the tested bacteria. This study is consistent with our results, which showed antimicrobial activity only against *Staphylococcus aureus*. The essential oil of *H. italicum* may have an inhibitory effect on some Gram-negative microorganisms, but it is probably necessary to find the ideal compound and concentration and extend the test to a wider range of microorganisms.

## 5. Conclusions

The GC-MS analysis of the eight fractions of *H. italicum* essential oil separated every 15 min during steam distillation revealed the significant differences between fractions. The results obtained show a significant influence of the distillation time on the changes in the composition of the essential oil. The content of monoterpenes and angelate decreases significantly during distillation, while the content of sesquiterpenes and sesquiterpene oxides increases towards the end of the distillation. In conclusion, a shorter distillation time will make the oil richer in monoterpenes and more suitable for the perfume and cosmetics industry. Although it showed moderate antimicrobial activity against *Staphylococcus aureus*, the *H. italicum* essential oil has significant potential for use as a preservative in the food and cosmetics industries. Further studies on the antimicrobial and antifungal properties of *H. italicum* essential oil should be continued and include other microbes and fungi such as *Aspergillus niger*. In the following studies, the most important components for specific applications in the perfume and cosmetic industry should be highlighted. Then, the distillation conditions can be adjusted according to our experiment to obtain an essential oil enriched with the desired components.

**Author Contributions:** Conceptualization, M.P. and S.B.; Methodology, M.P. and Z.M.; Software, M.P. and D.K.; Validation, M.P., S.B., Z.M. and D.K.; Formal Analysis, V.P., M.P., Z.M. and J.J.; Investigation, M.P., V.P. and J.J.; Data Curation, M.P., S.B., Z.M. and D.K.; Writing—Original Draft Preparation, V.P. and M.P.; Writing—Review and Editing, D.K. and T.P.; Visualization, V.P., M.P. and D.K.; Supervision, M.P. and D.K.; Funding acquisition, D.K. and T.P. All authors have read and agreed to the published version of the manuscript.

**Funding:** This study was carried out within the "Food Safety and Quality Center" (K.K.01.1.1.02.0004), a project funded by the European Regional Development Fund.

**Data Availability Statement:** The datasets presented in this article are not readily available because the data are part of an ongoing study.

**Conflicts of Interest:** Author Vesna Petrović was employed by the company Nicro d.o.o. Author Jerko Jukić was employed by the company Mahovina d.o.o. The remaining authors declare that the research was conducted in the absence of any commercial or financial relationships that could be construed as a potential conflict of interest.

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
