# Peer review of "Duration of Steam Distillation Affects Essential Oil Fractions in Immortelle (Helichrysum italicum)"

_horticulturae, doi:10.3390/horticulturae10020183_

Round 1

Reviewer 1 Report

Comments and Suggestions for Authors

Please see comments in the attached file.

Author Response

Dario Kremer

University of Zagreb

Faculty of Agriculture

Croatia

e-mail: dkremer@agr.hr

Zagreb, February, 9. 2024.

                Dear Reviewer 1,

Please find enclosed answers on your comments on manuscript "Variation of essential oil fractions of cultivated Helichrysum italicum (Roth) G. Don (Asteraceae) depending on the duration of steam distillation", with changed title: Duration of steam distillation affects essential oil fractions in Immortelle (Helichrysum italicum).

 List of questions and comments are given below. At first, thank to reviewer 1 for a carefully read text.TitleComment: Suggest to change the title to: Duration of steam distillation affects essential oil fractions in Immortelle Authors answer: The suggestion was accepted and the title was changed. AbstractComment: The abstract needs to be revised:1. Suggest to write the statement of the problem in the 1st sentence of the abstract. In other word, why you need to do this study?2. The treatments and conditions are not clear. In other word, the methodology is not clear. 3. The results are not described well.Authors answer: The abstract was revised and results are better described. Comment: Please mention which fraction showed the best results.Authors answer: The abstract was revised and results are better described. Comment: Line 25: showed instead of showAuthors answer: This part of the abstract was removed. KeywordsComment: Suggest to add "extract" and "cosmetic industry"Authors answer: New keywords "extract" and "cosmetic industry" were added. 1. IntroductionComment: The statement of problem is not clear. In other word, what is the current gap? and how your study is going to fill it. What is/are the implications of your study? Please elaborate more.Authors answer: This paragraph was changed. 2. Materials and MethodsComment: Please give a reference. Please clearly mention the treatments or fractions. Please mention the specification of each treatment. Only mentioning different fractions is not enough.Authors answer: There is no reference because we do that for the first time. The specification of each treatment was given. 3. ResultsComment: Please mention the units for these values given in the Table.Authors answer: We added unit for RT (min), while RI has no unit. Comment: What are RT and RI?Authors answer: RT and RI are mentioned for the first in the section Materials and methods. Comment: Your results are non-significant. It is contrary to your discussion in line 258-260: "If the distillation period is too long, the oil will become enriched with sesquiterpene oxides, which can have a negative effect on the fragrance of the essential oil".Authors answer: We cannot agree that the results are not significant. Based on our study, the distillation time can be adjusted and the negative effects mentioned in line 258-260 can be predicted. Comment: Please give a subheading. For example, 3.2Authors answer: subheading was added 4. DiscussionComment: Line 256: had instead of hasAuthors answer: changed according to reviewer’ suggestion 5. ConclusionsComment: Not a good conclusion. What are the implication of your study? What is your conclusive sentence? What is your suggestion for the future study?Authors answer: The section Conclusion is rewritten explaining the impact of our study and suggesting the future ones. Sincerely,                     Dario Kremer  

Reviewer 2 Report

Comments and Suggestions for Authors

The study of Marinko Petrović et al. aimed to analyze the essential oil fractions of cultivated Helichrysum italicum (Roth) G. Don (Asteraceae) using GC-MS. In addition, the antimicrobial activity of H. italicum essential oil was evaluated.

The manuscript is clear, informative to readers, and in accordance with the aim and scope of Horticulturae.

However, there is some information that needs to improve to make the manuscript more informative, including:

1. Figures about the antimicrobial activity of H. italicum essential oil against Staphylococcus aureus ATCC 25923 should add in the Supplemental material to enhance the information.

2. Not only antimicrobial but also antifungal properties have high potential for use as preservatives in the food and cosmetics industries. The antifungal properties against Aspergillus niger are particularly important in perfumes and cosmetics. I highly recommend further study of this activity for your future research.

Author Response

Dario Kremer

University of Zagreb

Faculty of Agriculture

Croatia

e-mail: dkremer@agr.hr

Zagreb, February, 9. 2024.

     Dear Reviewer 2,

 Please find enclosed answers on your comments on manuscript "Variation of essential oil fractions of cultivated Helichrysum italicum (Roth) G. Don (Asteraceae) depending on the duration of steam distillation", with changed title: Duration of steam distillation affects essential oil fractions in Immortelle (Helichrysum italicum).

List of questions and comments are given below. At first, thank to reviewer 1 for a carefully read text. Reviewer 2 Comment: Figures about the antimicrobial activity of H. italicum essential oil against Staphylococcus aureus ATCC 25923 should add in the Supplemental material to enhance the information.Authors answer: Figure was added. Comment: Not only antimicrobial but also antifungal properties have high potential for use as preservatives in the food and cosmetics industries. The antifungal properties against Aspergillus niger are particularly important in perfumes and cosmetics. I highly recommend further study of this activity for your future research.Authors answer: The suggestion was incorporated in the text.  Sincerely,                     Dario Kremer   

Reviewer 3 Report

Comments and Suggestions for Authors

[Horticulturae] Manuscript ID: horticulturae-2851893

Title: Variation in essential oil fractions of cultivated Helichrysum
italicum (Roth) G. Don (Asteraceae) depending on the duration of steam
distillation
Special Issue: Medicinal and Aromatic Plants: Isolation, Characterization and
Biological Activities

This study looked at the influence of distillation time on the composition of cultivated Helichrysum italicum essential oil. The work could be of interest to the reader. However, the work lacked in-depth discussion. The new aspect of this study was not fully explained.

Comments:

-The authors mentioned in the abstract that the study's findings are connected to the fragrance of essential oils. This is a good ending. There was no information or debate in the text.

-Line 64: What is the type of expanded application in the perfume and cosmetic industry?

-Lines 67–69: Please include the citation to this assertion.

-Line 89: Please modify "several days" to a precise number.

-Why did the authors utilize two types of gram-negative bacteria?

-Figure 2: Please added the chemical name to the graphic to help the reader understand. Specifically, the primary component of the essential oil.

-It needs to be clarified about the essential oil without fractionation. How should you prepare this essential oil? More details should be included.

-The authors should re-examine the data in Table 1 and the GC-MS spectrum since the strongest peak of the essential oil was observed at a retention time of 9.70 (Figure 2), while the table indicated that the highest value was detected at 6.29.

-Figure 3: The authors must explain any figure legends that are not utilized in this way.

-Checked the font and size of the text.

-Can the authors organize the findings into sections such as 3.1 GC-MS profile, 3.2 the association between distillation duration and chemical degradation, and 3.3 antibacterial activity?

-The description in the discussion section lacks clarity on antibacterial action, particularly F1 for gram-positive bacteria. There was no antibacterial action. What happened to this condition? Why did other microorganisms show no activity? Please have an in-depth discussion.

-How does this experiment relate to the perfume industry? The writers must provide the commentary in this section.

- The conclusion needs to be revised.

Comments on the Quality of English Language

English editing is necessary.

Author Response

Dario Kremer

University of Zagreb

Faculty of Agriculture

Croatia

e-mail: dkremer@agr.hr

Zagreb, February, 9. 2024.

                Dear Reviewer 3,

Please find enclosed answers on your comments on manuscript "Variation of essential oil fractions of cultivated Helichrysum italicum (Roth) G. Don (Asteraceae) depending on the duration of steam distillation", with changed title: Duration of steam distillation affects essential oil fractions in Immortelle (Helichrysum italicum).

List of questions and comments are given below.This study looked at the influence of distillation time on the composition of cultivated Helichrysum italicum essential oil. The work could be of interest to the reader. However, the work lacked in-depth discussion. The new aspect of this study was not fully explained.  Reviewer 3Comment: The authors mentioned in the abstract that the study's findings are connected to the fragrance of essential oils. This is a good ending. There was no information or debate in the text.Authors answer: In according to reviewer suggestions we add a few sentences in section Discussion.  Comment: Line 64: What is the type of expanded application in the perfume and cosmetic industry?Authors answer: We add a sentence in the mentioned place (lines 56-57). Comment: Lines 67–69: Please include the citation to this assertion.Authors answer: Citation is added (line 72). Comment: Line 89: Please modify "several days" to a precise number.Authors answer: it was done, i.e. "several days" was changed in two weeks. Comment: Why did the authors utilize two types of gram-negative bacteria?Authors answer: These are, relatively speaking, the usual microorganisms that we use when testing the antimicrobial properties of similar products. From past experience in testing consumer products (soap, disinfectants, wet wipes, wet disinfecting wipes...) the most common reason for product non-compliance was contamination with strains of Escherichia coli and Pseudomonas aeruginosa (or more often Pseudomonas spp.). Comment: Figure 2: Please added the chemical name to the graphic to help the reader understand. Specifically, the primary component of the essential oil.Authors answer: Due to the large number of components, it is not possible to put names on the figure. In our opinion, describing only the major components would not contribute to better understanding of composition. Details are precisely given in table 1. Comment: It needs to be clarified about the essential oil without fractionation. How should you prepare this essential oil? More details should be included.Authors answer: The preparation of essential oils was given more detailed. Moreover, the point of this study is to give an insight into how the distillation time will affect the proportion of components in immortelle essential oil, and it is up to the producer to decide what kind of essential oil that want to produce for intended purpose. Comment: The authors should re-examine the data in Table 1 and the GC-MS spectrum since the strongest peak of the essential oil was observed at a retention time of 9.70 (Figure 2), while the table indicated that the highest value was detected at 6.29.Authors answer: We would like to thanks to reviewer for carefully review. The figure was changed with new one. 

Comment: Figure 3: The authors must explain any figure legends that are not utilized in this way.

-Checked the font and size of the text.Authors answer: It was done. 

Comment: Can the authors organize the findings into sections such as 3.1 GC-MS profile, 3.2 the association between distillation duration and chemical degradation, and 3.3 antibacterial activity?

Authors answer: According to reviewer suggestion the section Results is divided in three subsections.

Comment: The description in the discussion section lacks clarity on antibacterial action, particularly F1 for gram-positive bacteria. There was no antibacterial action. What happened to this condition? Why did other microorganisms show no activity? Please have an in-depth discussion.

Authors answer: The discussion is deepening in terms of antibacterial action. Comment: How does this experiment relate to the perfume industry? The writers must provide the commentary in this section.Authors answer: Some relation between experiment and perfume industry was discussed. Comment: The conclusion needs to be revised.Authors answer: The section Conclusion is redesigned.  Sincerely, Dario Kremer  

Reviewer 4 Report

Comments and Suggestions for Authors

The manuscript titled ‘Variation in essential oil fractions of cultivated Helichrysum 2 italicum (Roth) G. Don (Asteraceae) with regard to the duration 3 of steam distillation’ provides a detailed description of the methodology, including the steam distillation process, GC-MS analyses, and antimicrobial analysis. In terms of significance, the findings of this study could be potentially valuable for the fields of horticulture.

Please find my comments and recommendations as follows.

Figure captions are a little too short and vague.

Figure 3A-D could be presented in a more presentable manner, also it would be nice if the caption had more detail on what A-D represents individually.

It would be valuable to include a section on future research directions, discussing potential follow-up studies or extensions of the current research.

Author Response

Dario Kremer

University of Zagreb

Faculty of Agriculture

Croatia

e-mail: dkremer@agr.hr

Zagreb, February, 9. 2024.

                Dear Reviewer 4,

Please find enclosed answers on your comments on manuscript "Variation of essential oil fractions of cultivated Helichrysum italicum (Roth) G. Don (Asteraceae) depending on the duration of steam distillation", with changed title: Duration of steam distillation affects essential oil fractions in Immortelle (Helichrysum italicum).

List of questions and comments are given below. At first, thank to reviewer 1 for a carefully read text. Reviewer 4The manuscript titled ‘Variation in essential oil fractions of cultivated Helichrysum 2 italicum (Roth) G. Don (Asteraceae) with regard to the duration 3 of steam distillation’ provides a detailed description of the methodology, including the steam distillation process, GC-MS analyses, and antimicrobial analysis. In terms of significance, the findings of this study could be potentially valuable for the fields of horticulture. Please find my comments and recommendations as follows. Comment: Figure captions are a little too short and vague.Authors answer: We try to better described figures. Comment: Figure 3A-D could be presented in a more presentable manner, also it would be nice if the caption had more detail on what A-D represents individually.Authors answer: We try to redesigned Figure 3. Comment: It would be valuable to include a section on future research directions, discussing potential follow-up studies or extensions of the current research.Authors answer: It was added in text. Sincerely,                     Dario Kremer   

Round 2

Reviewer 1 Report

Comments and Suggestions for Authors

The revised manuscript is improved according to reviewer's comment and this present form is suitable for publication.

Reviewer 3 Report

Comments and Suggestions for Authors

Thank you for revising the manuscript following the comments. 

Comments on the Quality of English Language

Minor editing of English language required.